# Efficacy of Rubella Vaccination after Co-Inoculation with Rhogam

**DOI:** 10.3390/v15091782

**Published:** 2023-08-22

**Authors:** Joshua S. Brunton, Regan N. Theiler, Ramila Mehta, Megan E. Branda, Elizabeth Ann L. Enninga, Vanessa E. Torbenson

**Affiliations:** 1Obstetrics Division, Department of Obstetrics and Gynecology, Mayo Clinic College of Medicine, 200 First Street SW, Rochester, MN 55905, USA; brunton.joshua@mayo.edu (J.S.B.); theiler.regan@mayo.edu (R.N.T.); 2Division of Quantitative Health Sciences, Mayo Clinic College of Medicine, Rochester, MN 55905, USA; mehta.ramila@mayo.edu (R.M.); branda.megan@mayo.edu (M.E.B.); 3Division of Research, Department of Obstetrics and Gynecology, Mayo Clinic College of Medicine, Rochester, MN 55905, USA; enninga.elizabethann@mayo.edu

**Keywords:** Rubella, Rhogam, MMR, post-partum vaccination

## Abstract

Congenital rubella syndrome is a constellation of birth defects that can have devastating consequences, impacting approximately 100,000 births worldwide each year. The incidence is much lower in countries that routinely vaccinate their population. In the US, postnatal immunization of susceptible women is an important epidemiological strategy for the prevention of rubella as the Center for Disease Control (CDC) does not recommend administering this vaccine during pregnancy due to its nature as a live attenuated virus vaccine. However, concerns that the co-administration of rubella vaccine with other immunoglobins (i.e., Rhogam) could compromise vaccine efficacy has produced warnings that can delay the administration of rubella vaccination postpartum, leaving women susceptible to the disease in subsequent pregnancies. We aimed to address whether the co-administration of the measles, mumps, and rubella (MMR) vaccine and Rhogam decreased antibody responses compared to those receiving only MMR vaccination. This retrospective cohort study utilized clinical data from 78 subjects who received the MMR vaccine and Rhogam after delivery and 45 subjects who received the MMR vaccine alone. Maternal demographics, pregnancy complications and rubella status at the start of a subsequent pregnancy were recorded for analysis. Overall, the two cohorts had similar baseline characteristics; however, lower parity was noted in the participants that received both MMR vaccination and Rhogam. Making assessments based on maternal antibody IgG index for rubella during the next pregnancy, we observed that 88% of the Rhogam + MMR vaccine group had positive serology scores, which was not significantly different from the 80% rate in the MMR-vaccine-only cohort (*p* = 0.2). In conclusion, no differences were observed in rubella immunity status in subsequent pregnancies in those mothers given both the MMR vaccine and Rhogam concurrently. Given these findings, warnings against co-administration of vaccines in combination with Rhogam appear unwarranted.

## 1. Introduction

Rubella virus, a *Rubivirus*, is an enveloped, positive-sense, single-stranded RNA virus that has a single antigenic type. By the time it was first isolated in 1962 [1], rubella infections had been well described and studied for nearly a century. Rubella transmission occurs via respiratory infection, where it initially replicates in the nasopharynx and regional lymph nodes. A rash is typically the first presenting symptom in younger children, while it usually presents after malaise, respiratory symptoms, and low-grade fever in older children and adults [2]. In 1941, after an epidemic of rubella infections, Norman Gregg described congenital cataracts in neonates born to mothers with rubella infections [3]. In pregnant patients with rubella infection, transplacental fetal infection may occur and tends to occur more frequently in the first trimester of pregnancy. A persistent fetal infection may lead to the disruption of cell division and destruction of fetal cells, which in turn can increase spontaneous abortions, stillbirths, and lead to a series of birth defects.

Congenital rubella syndrome (CRS) is the constellation of birth defects following rubella infection and can cause cataracts, hearing loss, mental retardation, congenital heart defects and growth restriction [4]. While the incidence in the United States is relatively low, CRS has been implicated in approximately 100,000 cases per year worldwide [5]. While currently uncommon, rubella epidemics can lead to drastic increases in morbidity for any given population and some countries, such as Japan, are seeing an increase in cases [5]. With effective vaccination policies, rubella could be eradicated since humans are the only known host. However, the lack of formal vaccination programs, vaccine hesitancy or refusal, and missed opportunities by the medical community for vaccine intervention can lead to lack of rubella immunity. An example of this is the 2019 measles outbreak in the United States (US), where a slight decline in vaccine uptake by several close-knit communities led to an outbreak despite the World Health Organization (WHO) having previously declared measles eliminated from the US [6]. As vaccine hesitancy increases in the wake of the COVID-19 pandemic [7], it has become increasingly important to educate patients and identify moments at which to intervene. One such clinical time point that lends itself as an optimal chance for vaccination intervention is during the postpartum period.

The current practice recommendation in the United States is to measure the rubella immunity of newly pregnant women and to vaccinate those without immunity with either the rubella vaccine or the measles, mumps, and rubella (MMR) vaccine postpartum, thereby decreasing the risk of rubella infection in a subsequent pregnancy. The Center for Disease Control (CDC) supports the administration of the MMR vaccine postpartum, although not during pregnancy due to MMR being an attenuated live virus vaccine [2]. An additional postpartum medication routinely administered is Rho(D) immunoglobulin (Rhogam). This is given to Rh-negative patients who have delivered a Rh-positive baby in order to decrease risk of alloimmunization and the development of hemolytic disease in a subsequent fetus. Approximately 15% of all deliveries require the administration of Rhogam, making this a commonly given medication on postpartum wards [8]. However, there has been concern that the postpartum administration of the rubella vaccine may generate suboptimal responses in rhesus (Rh)-factor-negative patients receiving the concomitant administration of Rhogam [9]. The concern arises from the possibility that administering a pooled immunoglobulin meant to suppress immune responses at the same time as a vaccine aimed at generating robust responses to viral antigens could theoretically blunt humoral and cellular immune responses.

For this reason, the CDC currently recommends checking for seroconversion several months after concomitant treatment in order to assure immunity [4]. This has resulted in several pharmacological warnings and occasional reluctance to co-administer these injections [9,10]. Small studies in the early 1970s indicated that co-administering Rhogam and rubella vaccine does not blunt the immune response to rubella [11,12]. In one study, 25 patients who were rubella-non-immune and Rh-negative received postpartum rubella vaccine and Rhogam and had the same seroconversion rates as a cohort of 21 rubella-non-immune patients receiving rubella vaccine alone. Even with the limited data supporting the coadministration of Rhogam and rubella vaccine, the warning to not administer the two concomitantly still exists. Therefore, we aimed to evaluate the serological outcomes of co-inoculating patients with both Rhogam and rubella vaccine postpartum. We hypothesized that rubella seroconversion rates will be similar between groups who received the MMR vaccine and Rhogam concurrently and the group receiving the MMR vaccine alone. Overall, our study aims to provide additional data regarding the safety and efficacy of co-inoculation to guide policy changes for improved evidence-based clinical care of pregnant patients.

## 2. Materials and Methods

This retrospective cohort study was performed within the Mayo Clinic Health System, which is a large network of hospitals serving Southeastern Minnesota. We identified all Rh-negative patients who delivered neonates between 1 January 2000 to 1 January 2021. This group was then further sub-classified into patients who had a rubella-non-immune or rubella-equivocal test result during their incident pregnancy. Our cohorts were created by then dividing our population into patients who received Rhogam due to a Rh-positive offspring and into those who did not receive Rhogam due to Rh-negative offspring. Each subgroup was then further queried to see if the subjects had a subsequent pregnancy during which a rubella antibody level was captured.

Inclusion criteria were women ≥ 18 years of age with Minnesota Research Authorization and a subsequent rubella titer captured during a later pregnancy. Chart review ensured that the timing of vaccinations was during the same postpartum hospitalization and that subsequent serology testing was available. Additionally, basic characteristics such as age, race and parity were collected for the incident pregnancy, follow-up rubella serology testing as well as the time interval between the administration of the MMR vaccine and Rhogam. Ethical committee approval (IRB 21-000667) was obtained prior to the initiation of this study and reviewing any records. All data were recorded in a secure REDCap database [13,14]. All serology testing was performed at our institutional laboratory. During the study timeframe, several different platforms were utilized including the Abbott AxSym Microparticle Enzyme Immunoassay (Abbott Laboratories, Abbott Park, IL, USA), BioMerieux VIDAS Enzyme Linked Fluorescent Assay (bioMérieux, Marcy-L’Etoile, France), SeraQuest ELISA (Awareness Technology, Palm City, FL, USA), and the BioRad BioPlex 2200 (Bio-Rad Laboratories, Hercules, CA, USA). Per our local laboratory guidelines, we utilized the antibody index values, which are semi-quantitative results generated during laboratory immunity assessment, with >1 being defined as a positive result, >0.7 and <1 as an equivocal result, and <0.7 defined as a negative result.

## 3. Statistical Analysis

Descriptive statistics of maternal features were reported as means and standard deviations or counts and percentages. Comparison between MMR-vaccine-only pregnancies versus MMR and Rhogam vaccine pregnancies were estimated with Kruskal–Wallis tests or chi-squared tests and defined as a *p* value ≤ 0.05. For the time interval between incident pregnancy and subsequent rubella serology testing, a generalized linear model adjusted by patient status and crossover status was conducted. SAS v9.4 (SAS, Cary, NC, USA) was used for analysis.

## 4. Results

During the study period, 44,076 unique deliveries occurred within our population, out of which 123 (0.28%) were identified as being eligible for study inclusion (Figure 1). In our cohort of 123 subjects, 78 subjects (53%) were identified to be Rh-negative via non-immune or equivocal rubella serology testing and had delivered a Rh-positive neonate, necessitating Rhogam administration. Forty-five subjects (37%) were found to be Rh-negative via non-immune or equivocal rubella serology testing and delivered a Rh-negative neonate, which did not require Rhogam. The maternal demographics of our cohort are displayed and compared in Table 1. The groups were balanced in terms of age at delivery, race, ethnicity, gestational age, and delivery modality. Subjects receiving rubella vaccination alone did not have a significantly higher parity than the group treated with the MMR vaccine plus Rhogam (*p* = 0.15). In total, 89.4% of the participants delivered at or after 37 weeks’ gestation (*p* = 0.39) and 74% had a vaginal delivery (*p* = 0.04). All participants received MMR vaccination during their postpartum hospitalization stay. One subject received Rhogam for a procedure 5 days prior to delivery, with the remainder receiving Rhogam and MMR within the same 24 h time period.

Subjects seeking prenatal care for their subsequent pregnancy had their rubella serology testing performed via a routine blood test during the first trimester (Table 2). In the group that previously received both MMR vaccination and Rhogam administration postpartum (n = 78), 69 were determined to be rubella-immune (88.5%). For the cohort receiving only MMR vaccination (n = 45), we observed that 36 subjects (80%) were seropositive for rubella, which was not significantly different from the number receiving the MMR vaccine + Rhogam (*p* = 0.20). Overall, 12% and 20% of subjects did not show rubella seroconversion in the MMR vaccine + Rhogam and MMR-vaccine-only groups, respectively. The time interval between the incident pregnancy and follow-up rubella serology testing was significantly different between groups (*p* = 0.02). Specifically, in those who did not seroconvert, the group receiving MMR vaccination + Rhogam had a longer interval (2.8 years) to follow-up titer measurements compared to 1.0 year for the MMR-vaccine-only group (Table 3).

## 5. Discussion

Our data demonstrate that patients receiving both Rhogam and the MMR vaccine during their postpartum hospitalization do not have worse rates of seroconversion into rubella immunity than those who received the MMR vaccine alone. These findings are consistent with those of smaller studies that found no difference between rubella titers in these two groups during a short time interval [11,12]. Our study evaluated the difference in real-world application after periods of months to years and found no effect of concomitant Rhogam and MMR vaccine administration on subsequent rubella immunity status.

Following vaccination, the introduced antigens stimulate multiple immune cell types. B cells, T cells and macrophages all respond to the proteins that have been introduced [15]. The macrophages fragment the proteins into antigenic pieces prior to major histocompatibility complex (MHC) formation, presenting them at the surface of the cell where they are recognized by T cells. The T cell recognition then stimulates B cells to begin producing antibodies to the recognized antigen fragments [16]. This immunological memory allows for the immune system to rapidly respond to infectious agents in subsequent exposures, thereby limiting the potential negative effects of those infections. After mounting an immune response to vaccination, a person’s humoral immunity can be assessed via measurement of IgG, IgA, IgM, and IgE if applicable. During pregnancy, the vertical transmission of immunoglobulin G (IgG) across the placenta confers both fetal and early postnatal protection [17].

The immune response of individuals is known to vary and the field of vaccinomics is just beginning to uncover some of the reasons certain sub-groups of patients respond differently to the same vaccine [18]. With contributing factors such as genetics, immune status, and medications being used, vaccine response is a complex and poorly understood area of study. While the administration of anti-D immunoglobin (Rhogam) has been suggested [9] to blunt the response to rubella vaccination, there are only small studies that have been completed to date on this issue. As more is understood regarding individual responses to vaccination, personalized approaches may be used in the future. However, as of now, that level of individualization does not occur.

With the increase in vaccine hesitancy following the COVID-19 pandemic [6,7], we believe that studies such as this one are important in demonstrating the effectiveness of co-administering two very necessary treatments to avoid both congenital rubella syndrome and alloimmunization complications in future pregnancies. By utilizing the postpartum period as a time point to educate and intervene, the medical community can continue to make strides toward the eradication of rubella. While extrapolation of this study’s results to other vaccines given in conjunction with Rhogam are outside of the current scope, it would be prudent to examine other immune responses to COVID-19, human papilloma virus (HPV), and varicella vaccination.

This study assessed the seroconversion of patients who received both Rhogam and MMR vaccination during their postpartum hospitalization. The comparison of rubella-non-immune, Rh-negative with Rh-negative neonate patients with the control group allowed us to demonstrate the lack of significant difference in the seroconversion rates of those who received postpartum vaccination with concomitant administration of Rhogam. The strengths of our study include a large and diverse patient population at a high volume academic medical center with an extensive clinical database of obstetric details. Even with a large population to query, the patients meeting our very specific criteria were limited. This is also to our knowledge the largest cohort used to analyze this clinical scenario and question to date. The main limitation of our design lies within the retrospective nature of our study. However, all patient data collection followed a strict protocol, and both clinical and laboratory records were retrieved from local, well-curated clinical databases. Another limitation lies within the fact that not all patients routinely had a subsequent pregnancy during that could be analyzed to identify whether they have undergone seroconversion from initial vaccination, which is not usually otherwise tested within this population. The final limitation is regarding the fact that varied rubella serology platforms were utilized throughout our study timeframe and how they mostly reported results in a qualitative manner, which did not allow us to compare the percentages of seroconversion across groups. However, this also most likely represents other institutions that have similarly adapted over time to new and emerging technology and testing equipment.

## 6. Conclusions

Rubella infection is a preventable pathogenic condition that can have severe consequences for a developing fetus and neonate. Due to hesitancy in administering the rubella vaccine and Rhogam concomitantly, our aim was to compare the seroconversion rates among patients receiving either the MMR vaccine + Rhogam versus the MMR vaccine alone. Given the findings in this study, i.e., that Rhogam administration at the time of MMR vaccination has no effect on subsequent rubella immunity status, further warnings regarding the coadministration of Rhogam with rubella vaccines appear unwarranted. This finding has implications for the mid-pregnancy and postpartum administration of other vaccines as well, including COVID-19, measles, mumps, varicella, and HPV vaccinations. Vaccinating susceptible patients postpartum decreases the risk of subsequent infections and improves outcomes for neonates, regardless of concomitant immunoglobulin administration. The incorporation of these findings into CDC recommendations is an important step toward changing hospital policy to have the greatest impact on patient outcomes.

## Figures and Tables

**Figure 1 viruses-15-01782-f001:**
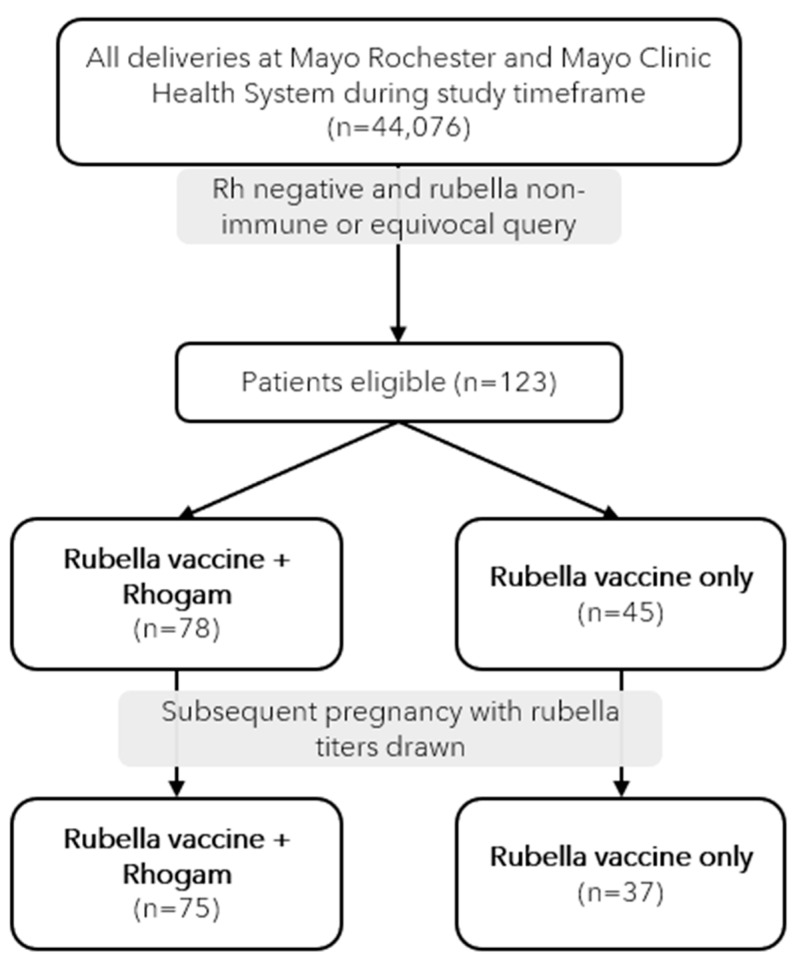
Study population selection and cohorts.

**Table 1 viruses-15-01782-t001:** Characteristics of participants who received MMR vaccination + Rhogam compared to MMR vaccination alone in incident pregnancy.

	Patient Type			
	MMR Vaccine + Rhogam	MMR Vaccine Only	Total	*p* Value
	(N = 78)	(N = 45)	(N = 123)	
Age at Delivery, Mean (SD)	27.4 (4.66)	28.0 (4.09)	27.6 (4.45)	0.43 ^1^
Race, n (%)				
on-White	5 (6.4%)	0 (0.0%)	5 (4.1%)	
White	73 (93.6%)	45 (100.0%)	118 (95.9%)	
Ethnicity, n (%)				
Hispanic or Latino	2 (2.6%)	0 (0.0%)	2 (1.6%)	
Not Hispanic or Latino	76 (97.4%)	45 (100.0%)	121 (98.4%)	
Para, n (%)				0.15 ^2^
0	15 (19.2%)	5 (11.1%)	20 (16.3%)	
1	39 (50.0%)	29 (64.4%)	68 (55.3%)	
2	18 (23.1%)	5 (11.1%)	23 (18.7%)	
3+	6 (7.7%)	6 (13.3%)	12 (9.8%)	
Gestational Age (weeks), n (%)				0.39 ^2^
37+	72 (92.3%)	38 (84.4%)	110 (89.4%)	
32–37	5 (6.4%)	6 (13.3%)	11 (8.9%)	
<32	1 (1.3%)	1 (2.2%)	2 (1.6%)	
Delivery Modality, n (%)				0.04 ^2^
Cesarean	19 (24.4%)	13 (28.9%)	32 (26.0%)	
Vaginal, Forceps or Vacuum	2 (2.6%)	6 (13.3%)	8 (6.5%)	
Vaginal, Spontaneous	57 (73.1%)	26 (57.8%)	83 (67.5%)	
Time interval between MMR and Rhogam (days), Mean, (SD)	0.68 (0.80)			

^1^ Kruskal-Wallis *p* value; ^2^ Chi-Square *p* value.

**Table 2 viruses-15-01782-t002:** Characteristics of participants who received MMR vaccination + Rhogam compared to MMR vaccination alone in their subsequent pregnancy.

	Patient Type			
	MMR Vaccine + Rhogam	MMR Vaccine Only	Total	*p* Value
	(N = 75)	(N = 37)	(N = 112)	
Age at Subsequent Delivery, Mean (SD)	29.8 (4.64)	29.6 (4.17)	29.7 (4.48)	0.74 ^1^
Race, n (%)				
Non-White	5 (6.7%)	0 (0.0%)	5 (4.5%)	
White	70 (93.3%)	37 (100.0%)	107 (95.5%)	
Ethnicity, n (%)				
Hispanic or Latino	2 (2.7%)	0 (0.0%)	2 (1.8%)	
Not Hispanic or Latino	73 (97.3%)	37 (100.0%)	110 (98.2%)	
Para, n (%)				0.52 ^2^
1	3 (4.0%)	1 (2.7%)	4 (3.6%)	
2	51 (68.0%)	30 (81.1%)	81 (72.3%)	
3	12 (16.0%)	4 (10.8%)	16 (14.3%)	
4+	9 (12.0%)	2 (5.4%)	11 (9.8%)	

^1^ Kruskal-Wallis *p* value; ^2^ Chi-Square *p* value.

**Table 3 viruses-15-01782-t003:** Crossover status of participants who received MMR vaccine + Rhogam compared to MMR vaccination alone and the interval of time between follow-up rubella titers.

	Patient Type		
	MMR Vaccine + Rhogam (N = 78)	MMR Vaccine Only (N = 45)	*p* Value
Rubella status @ 2nd pregnancy			0.20 ^1^
Negative/Equivocal	9 (11.5%)	9 (20%)	
Positive	69 (88.5%)	36 (80%)	
Time interval between incident pregnancy and subsequent rubella titer (years)			0.02 ^2^
Negative/Equivocal, Mean (SD)	2.8 (1.4)	1.0 (0.89)	
Positive, Mean (SD)	2.0 (1.3)	1.9 (1.1)	

^1^ Chi-Square *p* value; ^2^ F Statistic *p* value.

## Data Availability

The data presented in this study are available on request from the corresponding author. The data are not publicly available due to privacy concerns.

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
