# Peer review of "Efficacy of Rubella Vaccination after Co-Inoculation with Rhogam"

_viruses, 2023, doi:10.3390/v15091782_

Round 1

Reviewer 1 Report

Dear Editor

I have read with interest the manuscript submitted to me for review.

The reported results may have a certain interest for the evaluation of rubella vaccination and the association with Rhogam, but I believe that some clarifications must be made:

1. In a data collection lasting more than twenty years, the cases considered leave me a bit perplexed due to the excessive scarcity (especially for an important medical center such as the Mayo Clinic);

2. It is not clearly specified whether trivalent vaccine (MMR) or rubella vaccine alone was used;

3. The rubella vaccine can be inoculated during pregnancy in women with antibody titers below the protective values; Rhogam can be administered after childbirth, once Rh incompatibility has been established;

4. It would be interesting to know the antibody titre; unfortunately the values are provided as a ratio and therefore cannot be evaluated from this point of view (for example if there are differences between anti-rubella alone and anti-rubella + Rhogam);

5. Table 1: I guess C-section stands for cesarean delivery; it should be specified what is meant by vaginal, operational (use of forceps, use of a suction cup?).

Author Response

Reviewer 1

  1. In a data collection lasting more than twenty years, the cases considered leave me a bit perplexed due to the excessive scarcity (especially for an important medical center such as the Mayo Clinic);

Response: This represents a convenience sample of those cases with and without Rhogam administration that presented for a subsequent pregnancy within our health system and had research authorization. With the prevalence of Rh incompatibility approximately 15%, several cohort studies have shown about 9% of childbearing women to be rubella non-immune. These lower incidences couples with our inclusion criteria of a subsequent pregnancy with rubella titers quickly reduces our subjects eligible for inclusion. This was added in the limitations portion of the manuscript.

  1. It is not clearly specified whether trivalent vaccine (MMR) or rubella vaccine alone was used;

Response: In our study the trivalent vaccine MMR was used. We have added clarification in the introduction and discussion to make it more clear to the reader what vaccine was used in our study.

  1. The rubella vaccine can be inoculated during pregnancy in women with antibody titers below the protective values; Rhogam can be administered after childbirth, once Rh incompatibility has been established;

Response: As the Center for Disease Control (CDC) does not currently recommend MMR in pregnancy we have adhered at our institution to their guidelines. However, in an effort to make that clearer to reviewers and readers we have added to the introduction and included a citation with the CDC recommendation.

  1. It would be interesting to know the antibody titre; unfortunately the values are provided as a ratio and therefore cannot be evaluated from this point of view (for example if there are differences between anti-rubella alone and anti-rubella + Rhogam);

Response: Unfortunately, this is the case as the platform that was used throughout the study timeframe varied as the laboratory technology changed and was updated at our institution. We agree it would be interesting to be able to compare the titers/values across groups however we have mostly qualitative results with only semi-quantitative from the last decade of testing. We added this as an additional limitation of our study in the manuscript.

  1. Table 1: I guess C-section stands for cesarean delivery; it should be specified what is meant by vaginal, operational (use of forceps, use of a suction cup?).

Response: Added clarification of cesarean delivery.  Operative vaginal delivery has also been clarified into the use of forceps or vacuum to assist vaginal delivery.

Reviewer 2 Report

The Brief Report by Brunton et al. presents the result of a retrospective study comparing the serological response to rubella vaccine given post-partum to Rh-negative  women that were treated or not with anti-D immunoglobulin RhoGAM.  The results show that the percentage of seroconverted women at their subsequent pregnancy was similar in the two groups, suggesting that the administration of RhoGAM does not interfere with the efficacy of rubella vaccine.

Although this study has some limitations,  it is an interesting report that addresses with data the debate of whether immunoglobulins inhibit the immune response to a concomitantly administered vaccine.   

There are however a number of issues of unclear methodology and terminology, which  the authors should address:

1) Title: “co-inoculation” means that RhoGAM and rubella vaccine are administered at the same time, which does not seem to be the case (lines 96-98). I suggest to change this to “inoculation”, and, more importantly, to add the time interval between the two inoculations as an additional variable in the Tables (perhaps Table 3?) and determine if there is any significant relationship between seroconversion and interval before vaccination.

2) Line 100-107. The description of the testing methods is not informative and precise.  In line 100, I don’t think the author measured the titre of rubella IgG, but just the qualitative negative/equivocal/positive status. What is an “antibody index”? The authors should explain what platform (Bioplex 2200?) and test was used to measure rubella  IgG and why the cutoffs are set in the way they are.

3) On the same note, the authors identify an equivocal range, but there are no equivocal results presented in Table 3 or mentioned anywhere (except on line 90). Why not? Were there no equivocal results? Or  were they grouped with the negatives?

4) Line 107 – What is a “basic comparison”?  There are no data reported as means and SD in this manuscript, and therefore it is difficult to understand what is described here.

5) On the same note, I suggest to add 95% CI to the frequency data in the tables. Also, In Table 1 there are only 5 and 0 non Whites and 2 and 0 Latinos in the cells, which results in a meaningless chi-square analysis. I suggest to remove these comparisons from the Table and perhaps mention this in a note or a sentence in the text. Same goes for parity that should end at 3+, and gestational age with only a <32 group and an explanatory note for lower gestational ages.  What does 6/7 mean in Table 1?

6) Lines 121-122 – The authors state that “Nearly 90% […] had a spontaneous vaginal delivery”, but, according to Table 1, 32.5% did not.

7) Line 123-124 – Did the participants receive MMR(V) or a monovalent rubella vaccine.

8) Line 156-157 – Titres are not measured, or shown, in this study and therefore this sentence is confusing. Perhaps the authors mean “percentage of seroconversion”.  Actually, it would be informative to tease out semi-quantitative data from the rubella IgG results and correlate the Ab concentration in the two study groups, if possible.

9) Line 159-168 – I have several problems with this paragraph: a) this description ignores T-cell immunity, which is of great importance in clearing the primary infection and, likely, preventing re-infection. Although an Ab response is a good markers of immunization after a vaccine, it is not universally known if this is sufficient or even necessary for protection; b) therefore, I disagree with the statement that the goal of vaccination is simply  to induce humoral immunity; c) the authors refer multiple times to “pathogenicity of antigens”, but actually what is pathogenic is the virus or other infectious agents against which the vaccine is directed and not the antigen(s) which is (are) used for immunization. 

The authors should rephrase this paragraph.

10) Line 173 - …”administration of anti-D immunoglobulin”

11) Lines 209-210 – I suggest to add to this list measles and mumps vaccines.

Author Response

Reveiwer 2

1) Title: “co-inoculation” means that RhoGAM and rubella vaccine are administered at the same time, which does not seem to be the case (lines 96-98). I suggest to change this to “inoculation”, and, more importantly, to add the time interval between the two inoculations as an additional variable in the Tables (perhaps Table 3?) and determine if there is any significant relationship between seroconversion and interval before vaccination.

Response: We have added the time interval between administration to Table 1 which demonstrates that the majority of MMR vaccines and Rhogam were administered on the same day. We only had one patient with an interval greater than 72 hours.

2) Line 100-107. The description of the testing methods is not informative and precise.  In line 100, I don’t think the author measured the titre of rubella IgG, but just the qualitative negative/equivocal/positive status. What is an “antibody index”? The authors should explain what platform (Bioplex 2200?) and test was used to measure rubella  IgG and why the cutoffs are set in the way they are.

Response: Additional details regarding the testing methods and platforms utilized has been added to the material and methods section. The cutoffs are what is used by our institution to determine negative, equivocal, and positive.

3) On the same note, the authors identify an equivocal range, but there are no equivocal results presented in Table 3 or mentioned anywhere (except on line 90). Why not? Were there no equivocal results? Or  were they grouped with the negatives?

Response: Equivocal were grouped with negative as stated in line 90.  We have attempted to make this more clear by referring to the group as negative/equivocal in Table 3.

4) Line 107 – What is a “basic comparison”?  There are no data reported as means and SD in this manuscript, and therefore it is difficult to understand what is described here.

Response: Thank you for identifying this.  The statistical analysis section has been updated to clarify the analysis that was completed on lines 107-110.

5) On the same note, I suggest to add 95% CI to the frequency data in the tables. Also, In Table 1 there are only 5 and 0 non Whites and 2 and 0 Latinos in the cells, which results in a meaningless chi-square analysis. I suggest to remove these comparisons from the Table and perhaps mention this in a note or a sentence in the text. Same goes for parity that should end at 3+, and gestational age with only a <32 group and an explanatory note for lower gestational ages.  What does 6/7 mean in Table 1?

Response: Thank you for pointing out these issues.  We have updated Table 1 to remove the p-values with the 0 groups and have regrouped the variables noted.  For the 6/7, this means that the gestational age was 6 days (32 6/7 is 32 weeks and 6 days).

6) Lines 121-122 – The authors state that “Nearly 90% […] had a spontaneous vaginal delivery”, but, according to Table 1, 32.5% did not.

Response: Agree, we have added the more precise percentage that had a vaginal delivery.

7) Line 123-124 – Did the participants receive MMR(V) or a monovalent rubella vaccine.

Response:  In our study the trivalent vaccine MMR was used. We have added clarification in the introduction and discussion to make it more clear to the reader what vaccine was used in our study throughout the paper.

8) Line 156-157 – Titers are not measured, or shown, in this study and therefore this sentence is confusing. Perhaps the authors mean “percentage of seroconversion”.  Actually, it would be informative to tease out semi-quantitative data from the rubella IgG results and correlate the Ab concentration in the two study groups, if possible.

Response:  As most of the assays used during our study period were qualitative we are not able to tease out this data for our entire population. The sentence causing confusion was removed as we do not think it added to our primary findings.

9) Line 159-168 – I have several problems with this paragraph: a) this description ignores T-cell immunity, which is of great importance in clearing the primary infection and, likely, preventing re-infection. Although an Ab response is a good markers of immunization after a vaccine, it is not universally known if this is sufficient or even necessary for protection; b) therefore, I disagree with the statement that the goal of vaccination is simply  to induce humoral immunity; c) the authors refer multiple times to “pathogenicity of antigens”, but actually what is pathogenic is the virus or other infectious agents against which the vaccine is directed and not the antigen(s) which is (are) used for immunization. 

The authors should rephrase this paragraph.

Response: Rewrote paragraph to better explain in brief the immunologic mechanism behind vaccination and better differentiate between antigens and infectious agents as the reviewer recommended.

10) Line 173 - …”administration of anti-D immunoglobulin”

Response: Thank you for this correction. The appropriate changes have been made in the manuscript.

11) Lines 209-210 – I suggest to add to this list measles and mumps vaccines.

Response: Thank you for this feedback. We have added this to the manuscript.

Round 2

Reviewer 1 Report

The authors reply completely to my suggestions. I have not further comments.